# Language Models for Fall Risk Assessment in Children with Cerebral Palsy using Electronic Medical Records

Thasina Tabashum
*Computer Science and Engineering*
*University of North Texas*
Denton, TX, USA
thasinatabashum@my.unt.edu

Shou-Jen Wang
*Computer Science and Engineering*
*University of North Texas*
Denton, TX, USA
shou-jenwang@my.unt.edu

Joseph J. Krzak
*Shriners Children's*
Chicago, IL, USA
jkrzak@midwestern.edu

Karen M Kruger
*Shriners Children's*
Chicago, IL, USA
karen.kruger@marquette.edu

Adam Graf
*Shriners Children's*
Chicago, IL, USA
agraf@shrinenet.org

Ross S. Chafetz
*Shriners Children's*
Philadelphia, PA, USA
rchafetz@shrinenet.org

Jon R. Davids
*Shriners Children's*
Sacramento, CA, USA
jdavids@shrinenet.org

Anita Bagley
*Shriners Children's*
Sacramento, CA, USA
abagley@shrinenet.org

Jason Woloff
*Shriners Children's*
Tampa, FL, USA
ssienko@shrinenet.org

Susan E. Sienko
*Shriners Children's*
Portland, OR, USA
ssienko@shrinenet.org

Jeremy Bauer
*Shriners Children's*
Portland, OR, USA
jpbauer@shrinenet.org

Mark V. Albert
*Computer Science and Engineering*
*Biomedical Engineering*
*University of North Texas*
Denton, TX, USA
mark.albert@unt.edu

*Abstract*—Children with Cerebral Palsy (CP) face a heightened risk of falls, complicating treatment outcomes. Traditional manual scoring methods like the Cummings Fall Assessment Score are subjective and labor-intensive due to the diverse characterization of CP. Leveraging electronic medical records (EMRs) and advanced language models (LMs) offers a data-driven alternative for fall risk assessment. Because CP is a varied heterogeneous cohort LMs have not been thoroughly applied to assess fall risk. To address this, we utilized unstructured EMR data from 1,604 patients with CP from the Shriners Children's Hospital Network, employing Clinical BioBERT, BioBERT, and BERT$_{BASE}$ to predict fall risk. We explored two approaches: continued pre-training followed by fine-tuning with labeled data, and direct fine-tuning with supervised labeled data. Our findings indicate that continued pre-training does not guarantee performance improvements on downstream tasks, relative to only fine-tuning. This reduces the need for an extensive pre-training process. Only fine-tuned models were able to achieve a 0.71 F1 in prediction fall assessment risk, and 0.74 F1 scores when we excluded the borderline fall assessment score during training. The best performance achieved by Clinical BioBERT is a recall of 0.72 and a specificity of 0.80. Furthermore, CP is a complex, multifaceted condition often involving lengthy clinical notes that exceed 30,000 characters, which many LMs cannot process in a single context window. We propose a process where notes are assessed as a group of samples that fit the context window. Then a collective decision is made by probability weighted majority voting (PWMV). This approach improved model performance by 1% to 3% and demonstrated its effectiveness in enhancing fall risk prediction for children with CP. Our work lays the groundwork for evidence-based, data-driven treatment planning in pediatric CP clinical practice and research, significantly improving the efficiency and accuracy of CP patient care.

*Index Terms*—Cerebral Palsy; BERT; Fall Prediction; Language Model, Electronic Medical Records.

## I. INTRODUCTION

Cerebral palsy (CP) is a lifelong movement disorder that affects individual muscle control. The severity and specific symptoms of CP vary widely among individuals [1]. One major concern among individuals with CP is falls, previous studies showed that children with CP are more prone to falls [2]. Falls represent a significant cause of death and injury in pediatric hospitals, accounting for 25% to 52% of all treated child injuries [3]. Furthermore, falls can result in injuries that deteriorate the subject's condition. According to Jamerson et al., over half of the children who fell were not recognized as high-risk before the incident occurred [4]. This points to a gap in the fall risk assessment process, and standardized fall assessment tools can assist clinicians in evaluating the individual's conditions.

Assessment is important in CP management, including diagnosis, progress monitoring, quality of life, social skills, motor abilities, and more [5]. Continuous re-intervention and re-evaluation are needed for the dynamic nature of CP, as a result maintaining consistent assessment by different clinicians is challenging [6], [7]. To assess individuals with CP, clinicians maintain a large amount of documentation that can be found in electronic medical records (EMR). EMR contains rich information about subjects growing at a fast rate in clinical practice [8]. Traditional practices can be significantly improved with the integration of artificial intelligence (AI) and EMR [9], [10]. As recent advancements in language modeling in numerous domains showed promising results, these language models (LMs) can be valuable in incorporating significant information on the subject's overall condition to develop fall risk assessment tools.

One major transformer-based [11] LM is Bidirectional Encoder Representations from Transformers (BERT) [12] that showed promising results in the different avenues of natural language tasks such as question answering, natural language inference, and translation. BERT has also gained popularity in the medical domain, for instance, BioBERT is trained on medical literature [13], and performed well in medical entity recognition, question answering, and relation extraction. BERT also presented an ideal performance on longitudinal structured EMR data, such as Med-BERT [14], BEHRT [15], CEHR-BERT [16], and HiBERT [17]. A specialized clinical model on unstructured EMR data has also been proposed Clinical BioBERT [18] on intensive care unit (ICU) datasets, MIMIC [19]. Previously, BERT-based inpatient fall detection is proposed using Alberta hospital notes [20]. Although there are significant advancements in LMs, no previous LM addresses CP concerns. To address this limited population need, we utilized unstructured EMR data of 1,604 CP subjects from Shriners Children's Hospital Network and trained three BERT models, Clinical BioBERT, BioBERT, and BERT$_{BASE}$ on fall risk assessment. To our knowledge, no study has utilized LM on unstructured EMR data of CP subjects in rehabilitation settings to assess fall risk.

LMs require a significant amount of computation power and money, generally, accounting for two steps, pre-training and fine-tuning [21]. For instance, it took 23 days using 8 NVIDIA V100 (32 GB) GPUs to pre-train BioBERT [13]. Pre-training is self-supervised training without a downstream task label. As the domain narrows, continued pre-training is general practice [14], [20]. The two main objective functions for pre-training are masked language modeling (MLM) and next sentence prediction (NSP). Fine-tuning is a training model with labeled data. Pre-training is not always essential when substantial labeled data is available. Zoph et al. demonstrated that high-quality labeled data pre-training decreases accuracy [22]. With a robust and diverse dataset, models can often learn effectively without pre-training. If one can bypass the pre-training phase, it is possible to eliminate a computationally intensive step. However, it is crucial to investigate whether this omission impacts model performance. To analyze the necessity

of pre-training in our settings, we demonstrated that extensive pre-training is not always required when high-quality labeled data is accessible.

In addition one of the biggest challenges for LMs is the context window. The context window is the amount of text a model can process at a time. LMs generally have pre-defined context window sizes. Even large LMs with billions of parameters, for example, LLaMA can process a maximum of 2048 tokens [23]. Clinical notes are much longer than the context window. Training models with longer context windows requires a significant amount of computation power and time and increases operational costs. For example, Chen et al. demonstrated that further pre-training Transformers for 1000 batches the model only increased the context window from 2048 to 2560 [24]. Therefore, increasing the context window is not always computationally efficient. To overcome the context window challenge and maximize the information, we utilized confidence weighted majority voting (CWMV) [25] with the extension of the equality affect parameter proposed by Meyen et al [26]. We integrated CWMV into our settings using the model's probability classification layer output to calculate the weights. This approach is straightforward, easily adaptable, and computationally efficient.

In summary, we conducted a comprehensive comparison between pre-trained and non-pre-trained Clinical BioBERT, BioBERT, and BERT$_{BASE}$ in predicting fall risk assessment risk in CP. To our knowledge, this work is the first to explore the application of LMs for CP and, more specifically, for fall risk assessment. Non-pre-trained models can predict fall risks effectively, highlighting that general LMs, when fine-tuned appropriately, can be robust tools for healthcare applications. Moreover, to mitigate the limitation of the context window we utilized a probability weighted majority voting (PWMV) strategy to make collective decisions across multiple segments of text. PWMV allows us to combine predictions from different text segments in a way that gives more weight to certain positions, ensuring that important information is not lost due to context window constraints. By using PWMV, we improved the overall model performance by 1-3%. In this study, we propose a robust and effective strategy with valuable insights and well-validated generalized models with real-life hospital data on the CP cohort to predict fall risk.

## II. DATASET

### A. Pre-training Dataset

We have a total of $826,138$ EMR notes from Shriners Children's Hospital Network, it includes data from 23 different hospitals across a diverse set of locations, including Texas, Florida, California, Philadelphia, Ohio, Hawaii, Kentucky, Illinois, Washington, Utah, Massachusetts, South Carolina, Minnesota, Oregon, and Canada, encompassing 710 note types from $1,604$ subjects. This research was approved by the Institutional Review Board of the University of North Texas and the Shriners Children's Hospital System. We filtered the note types by removing those that were not present in at least 50 subjects. Next, we removed notes with fewer than

300 characters, we retained $670,857$ notes covering 275 note types. Table I illustrates the top 10 note types that almost all subjects have. Table II lists the after-thresholding minimum characters of the dropped note types. This note length varies from 300 characters to more than 30,000 characters. The text length distribution for the pre-training notes is depicted in Fig 1a.

TABLE I: Top 10 most frequent note type.

| Top 10 most frequent note type | # of Subjects |
|---|---|
| Coding Summary | 1601 |
| Outpatient Intake Information PowerForms | 1601 |
| Outpatient Progress Note | 1588 |
| Report | 1576 |
| Operating Room Nursing Record | 1566 |
| Admission History PowerForms | 1552 |
| Discharge Instructions PowerForms | 1539 |
| Patient Education PowerForms | 1539 |
| Telephone Triage PowerForms | 1537 |
| Discharge Follow-up PowerForms | 1533 |

TABLE II: Note types disregarded because they did not meet the required character length.

| Thresold | Note Types Dropped | Total Notes |
|---|---|---|
| 150 | Religious Preference, Double click to view, Language Indicated at Registration, Admitted From, County of Residence | 741,296 |
| 200 | Telehealth Consent ST, Legal Guardian | 715,763 |
| 250 | Precautions RTF | 694,199 |
| 300 | Planned Discharge Dispositions | 670,857 |

### B. Downstream Task Dataset

For the fall risk assessment predictions task, we used the Cummings Fall Assessment Score. Cummings Fall Assessment Score has 6 components: history of fall, physical alterations/impairments, equipment, functional status, cognitive/psychological, and medications that alter equilibrium. Each component is scored by clinicians. Table III describes each component of the total Cummings Fall Assessment Score. If the total score is 8 or more that is considered high-risk, 0 is considered no-risk, otherwise low-risk. For this study, we considered 0-7 as low-risk and 8-16 as high-risk. The threshold between 7 and 8 is practiced by this hospital network. We had $11,213$ samples from $1,144$ subjects that include the Cummings Fall Assessment Score, encompassing 14 different note types. Fig. 1b describes the lengths of these $11,213$ samples vary, ranging from 525 to $24,589$ characters.

We listed the note types with the number of subjects and samples in Table IV. Subjects can be assessed multiple times throughout treatment. One subject might not have the same fall score throughout their treatment. We showed score distribution by the number of subjects in Fig 2. The occurrence of fall samples varies from subject to subject. The time intervals between the documented fall scores Fig 3 illustrates the number of samples between the second week to the 61st week time intervals. The first week has the most samples that is 5026.

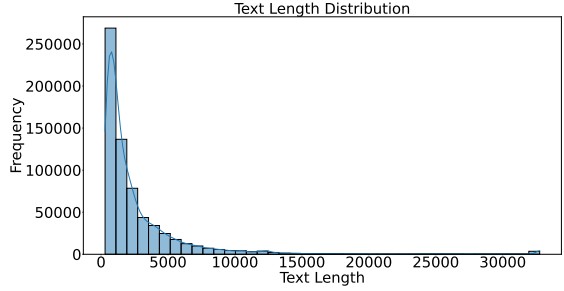

(a) Pre-training text length distribution.

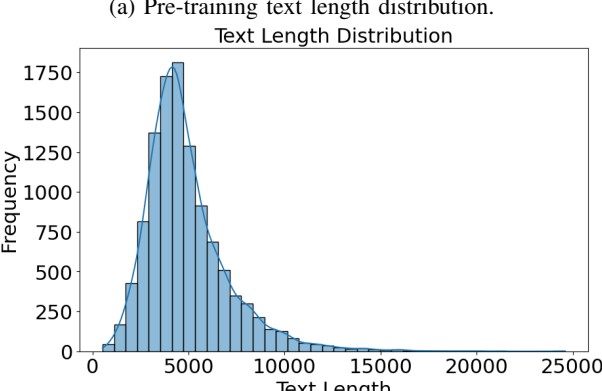

(b) Downstream task: Fall notes text length distribution.

Fig. 1: Distribution of text lengths and their corresponding frequencies across the dataset.

TABLE III: Components of Cummings Fall Assessment Scores.

| Components | Description | Score |
|---|---|---|
| History of Fall | Immediate or within 3 months | 0=No, 3=Yes |
| Physical Alterations/ Impairments | Surgery within admission, underlying medical conditions | 0=No, 3=Yes |
| Equipment | IV/Heparin Lock, IV Pole, or Foley Catheter | 0=No, 2=Yes |
| Functional Status | Altered Mobility Gait/Transferring Problems. Significant Metabolic Disturbances (Hypotension, hypoxia, hypovolemia), Use of Ambulatory Aids | 0=No, 1=weak 2=Impaired 3=Crutches, walker, brace |
| Cognitive/ Psychosocial | Impaired Mental Status Developments Delay, Bio-Behavioral Concerns, (ADHD,depression, oppositional defiant behavior) | 0=No, 2=Neuro Limitations due to illness; Bio-Behavioral concerns |
| Medications that Alter Equilibrium | Narcotics, Anti-Convulsive, Antipsychotics, Sedatives, Laxatives, Chemotherapy, or Hypotensive Meds | 0=No, 3=Yes |

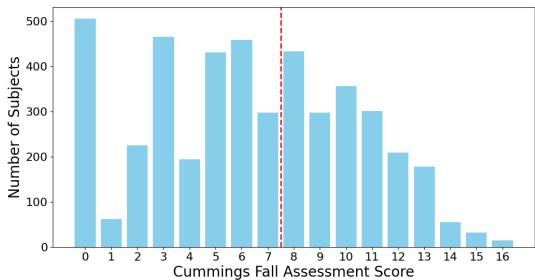

Fig. 2: Cummings Fall Assessment Score distribution by subjects, indicating low-risk (0-7) and high-risk (8+).

TABLE IV: Breakdown of note types on downstream task by the number of subjects and notes.

| Note Types | # of Subjects | # of Samples |
|---|---|---|
| Cast Application PowerForms | 20 | 24 |
| PT Evaluation/Treatment Note PowerForms | 33 | 41 |
| Basic Admission Information PowerForms | 34 | 36 |
| Discharge Instructions PowerForms | 35 | 40 |
| Pre-procedure Checklist PowerForms | 41 | 54 |
| Pain Management Catheter Care PowerForms | 44 | 77 |
| PACU Assessment Powerforms | 50 | 76 |
| Admission History PowerForms | 104 | 115 |
| Postprocedure Assessment PowerForms | 186 | 375 |
| Patient Summary, Outpatient | 193 | 718 |
| Patient Education PowerForms | 238 | 322 |
| Ongoing Shift Assessment PowerForms | 280 | 2972 |
| Outpatient Intake Information PowerForms | 523 | 3846 |
| Admission Assessment PowerForms | 1049 | 2517 |

PT: Physical Therapy, PACU: Post Anesthesia Care Unit.

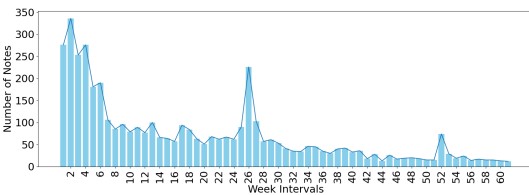

Fig. 3: Note counts per week by subjects: from 2nd to 61st week.

## III. METHOD

Our approach for assessing fall risk assessment using clinical notes is outlined in Fig 4. This process consists of three (without pre-training)/four (pre-training) main stages: section segmentation of each clinical note to fit the context window, pre-training BERT or non-pre-trained, fine-tuning, and collective decision-making.

### A. Data Pre-processing

For pre-training, we removed fall scores. We split all notes following Clinical BioBERT [18] pre-processing steps, and Scispacy [27] is used to perform sentence extraction. After that, we have 12,067,470 samples for pre-training. Similarly, we performed section extraction for fall risk assessment prediction. Each note is addressed as one sample for fall risk assessment prediction, and portions that fit the context window from one note are addressed as sections. We have $423,188$

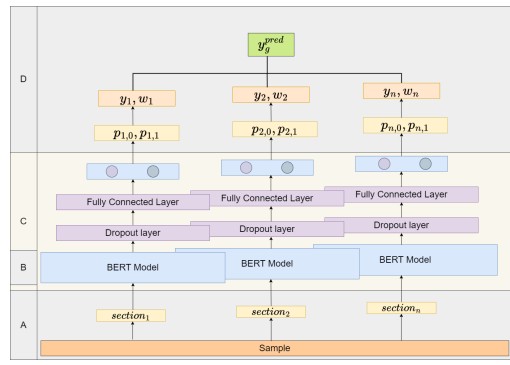

Fig. 4: General overview of our approach: Three/Four Stage Workflow, and each section from one sample is independent during training: A) Data Pre-processing, B) Pre-training, C) Fine-tuning, and D) Probability Weighted Majority Voting.

sections from $11,213$ samples to predict fall risk assessment. The mean sections from one sample is 38 ($38.12 \pm 16.83$). The average number of samples by subject is $9.8 \pm 11.55$.

### B. Pre-training BERT

We used the MLM objective function for pre-training and pre-trained Clinical BioBERT, BioBERT, and BERT$_{\text{BASE}}$ in the same setup. We pre-trained all the models with a 15% mask percentage. Following the original BERT [12] masking strategy, we replaced the i-th token with the mask token 80% of the time, a random token 10% of the time, and unchanged 10% of the time. Following the Clinical BioBERT [18], we fixed other hyper-parameters such as the duplication factor set to 5. We used 128 max length for the tokenizer with padding of maximum length, and setting truncation true. The batch size is 32. The learning rate is $5 \times 10^{-5}$ with 10,000 warmup steps. The weight decay is 0.01. The models are pre-trained to predict the original token with cross-entropy loss. We achieved a satisfactory loss on the test set within 31,562 steps. We used Adam Optimizer.

### C. Fine-tuning BERT

For fine-tuning BERT models we used the same hyper-parameters for all models. For classification, we added a dropout layer with 0.1, and fully connected layers with 128 dimensions with Relu activation. The learning rate was set $1 \times 10^{-5}$, batch size 32. For classification, softmax is used. The models were fine-tuned using 32 batch sizes for 2 epochs. During training individual sections were treated independently.

### D. Probability Weighted Majority Voting

To aggregate the predictions from sections from one sample, we used probability weighted majority voting (PWMV) following Meyen et al. strategy [26]. The individual weight is calculated using equation 2 which is logarithmic odds. Here, individual output probabilities are $p_i$. $max(p_i)$ is the maximum probability output of the classification layer between two classes. Aggregating these weights $w_i$ and $y_i$, group output $y_q^{PWMV}$ that is collective output for all the sections from one

sample is calculated using equation 3. $\beta$ is the equality effect that weighs individual votes. Here $\beta$ can be between zero to infinity, $\beta \in [0, \infty]$. if $\beta = 1$ the weights become unchanged which will be naive PWMV. On the contrary, if $\beta = 0$, every weight becomes 1 which results in majority voting (MV). The final group prediction is decided based on equation 4.

$$y_i = \begin{cases} 1, & \text{if } \arg\max(p_i) = 1 \\ -1, & \text{otherwise} \end{cases} \quad (1)$$

$$w_i = \log\left(\frac{max(p_i)}{1 - max(p_i)}\right) \quad (2)$$

$$y_q^{PWMV} = \text{sign}\left(\sum_{i=1}^{n} w_i^{\beta} y_i\right) \quad (3)$$

$$y_g^{pred} = \begin{cases} 1, & \text{if } y_q^{PWMV} >= 0 \\ 0, & \text{otherwise} \end{cases} \quad (4)$$

### E. Model Training and Evaluation

Every subject in the CP cohort exhibits unique characteristics and responses to treatment. To ensure the model's generalizability and robustness, subject-wise testing is crucial [28], [29]. To properly validate the model, we tested by subject. For pre-training, we used a test set of 20% subjects. Next for the fall risk assessment prediction, the training set is split 80% (n=916) subjects, and 20% (n=288). The test subjects were excluded from pre-training and fine-tuning. The total sections for the fall risk assessments are 342,616 in the training set and 80,572 in the test set. In the training set the number of low-risk is $220,963$ and high-risk is 121,653. For evaluation, each model was fine-tuned using the training set, and all models were tested in the same separate test set of 288 subjects.

## IV. RESULTS

### A. Model Pre-training Results

We pre-trained ClinicalBioBERT, BioBERT, and BERT$_{BASE}$ for 31,562 steps and 18,750 steps. We evaluated the pre-trained models by MLM loss on the test set. There is not a significant difference between 18k to 31k steps. The losses of 31,562 steps on the test set were 0.235, 0.241, and 0.245; the losses on the training set were 0.107, 0.119, and 0.110 for ClinicalBioBERT, BioBERT, and BERT$_{BASE}$ respectively. The loss until 18K steps is depicted in Fig 5. The results indicate that all models demonstrate comparable performance on the MLM objective. Initially, BioBERT underperformed compared to the other models. However, by the 3,000-step mark, all models achieved similar performance levels.

### B. Fine-tuning Results

*1) $\beta$ search:* To aggregate the decision from individual sections, we used equation 3 to make one decision per sample. We grid search $\beta$ for each model in the training set from 0 to 4 with the step size of 0.01. The $\beta$ scores are listed in table V. For comparison, we also applied $\beta = 0$ which is majority

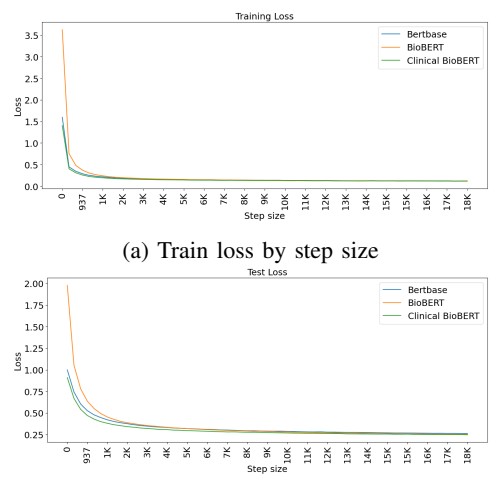

(a) Train loss by step size

(b) Test loss by step size

Fig. 5: Pre-training train and test loss across step sizes.

voting (MV), and $\beta = 1$ that Naive PWMV. The larger the $\beta$ the greater the impact of high confidence scores relative to low confidence scores. Our results show that all the models have larger $\beta$ values.

TABLE V: $\beta$ values

| Model | Training | $\beta$ |
|---|---|---|
| ClinicalBioBERT | Pre-trained | 2.98 |
| BioBERT | Pre-trained | 1.90 |
| BERT$_{BASE}$ | Pre-trained | 2.90 |
| ClinicalBioBERT | Non-Pre-trained | 3.90 |
| BioBERT | Non-Pre-trained | 3.90 |
| BERT$_{BASE}$ | Non-Pre-trained | 1.96 |

*2) Prediction Results:* The evaluation metrics F-1, precision, and recall on the test set are shown in Table VI. Comparing the results Clinical BioBERT and BERT$_{BASE}$ decrease performance when pre-trained, whereas BioBERT improves after pre-training. PWMV ($\beta > 1$) gave the best results. The best F1 score is 0.71 achieved by non-pre-trained Clinical BioBERT and pre-trained BioBERT using PWMV ($\beta > 1$) . Pre-trained BioBERT with PWMV ($\beta = 2.90$) achieved the best recall of 0.73. The specificity of all the models is more than 0.77. The specificity Clinical BioBERT achieved is 0.80.

To analyze model performance on different Cummings Fall Assessment Scores, we showed the confusion matrix of each score 0 to 16 in Fig 6 with recall values on the top of the confusion matrix. The recall values decrease for the 7 and 8 scores which are the thresholds between high and low-risk. The recall value for 7, and 8 is only 0.5. As illustrated in Fig 6, the model can predict better if subjects are further away from the threshold. The difference between 7 and 8 is considerably hard to learn. To investigate the effect of borderline samples (7 and 8), we fine-tuned the model without 7 and 8 scores and compared the results on all testing samples (including 7 and 8). The model's performance improved significantly after we excluded borderline samples. The F1 score increased

TABLE VI: Evaluation of Clinical BioBERT, BioBERT, and BERT$_{BASE}$ models on test set with PWMV Method comparing with $\beta$ =0 (MV), 1 (Naive PWMV). Best scores are bolded.

| Training | Model | F1 | Recall | Precision |
|---|---|---|---|---|
| **Clinical BioBERT** | | | | |
| Pre-trained | Individual | 0.66 | 0.63 | 0.69 |
| | MV | 0.67 | 0.64 | 0.69 |
| | Naive PWMV | 0.67 | 0.64 | 0.69 |
| | PWMV | 0.67 | 0.66 | 0.69 |
| Non-Pre-trained | Individual | 0.69 | 0.68 | **0.70** |
| | MV | 0.70 | 0.69 | **0.70** |
| | Naive PWMV | 0.70 | 0.70 | **0.70** |
| | PWMV | **0.71** | 0.72 | **0.70** |
| **BioBERT** | | | | |
| Pre-trained | Individual | 0.68 | 0.70 | 0.67 |
| | MV | 0.70 | 0.71 | 0.69 |
| | Naive PWMV | 0.70 | 0.72 | 0.68 |
| | PWMV | **0.71** | **0.73** | 0.68 |
| Non-Pre-trained | Individual | 0.67 | 0.65 | 0.69 |
| | MV | 0.69 | 0.68 | **0.70** |
| | Naive PWMV | 0.69 | 0.68 | **0.70** |
| | PWMV | 0.69 | 0.67 | **0.70** |
| **BERTBase** | | | | |
| Pre-trained | Individual | 0.66 | 0.65 | 0.68 |
| | MV | 0.67 | 0.65 | 0.69 |
| | Naive PWMV | 0.67 | 0.65 | 0.69 |
| | PWMV | 0.67 | 0.65 | 0.69 |
| Non-Pre-trained | Individual | 0.68 | 0.68 | 0.68 |
| | MV | 0.69 | 0.69 | 0.69 |
| | Naive PWMV | 0.69 | 0.69 | 0.69 |
| | PWMV | 0.69 | 0.69 | 0.69 |

from 0.71 to 0.74, and the recall improved from 0.72 to 0.77. Further testing without these borderline samples showed an increase in the F1 score from 0.74 to 0.78, and recall rose from 0.74 to 0.80. This suggests that the borderline samples might lack sufficient certainty, impacting the model's performance. Clinicians might also find it challenging to distinguish between cases labeled as 7 and 8. This leads to uncertainty in true labels due to the inherent ambiguity in these cases.

Moreover, we evaluated the non-pre-trained Clinical BioBERT with PWMV ($\beta = 3.90$) in a smaller subset of training subjects. We decreased 50% subjects in the training set and listed the evaluation metrics in Table VII. As the number of subject data increases the model performance can be increased.

TABLE VII: Performance metrics comparison with reduced training set size.

| # of Subjects | F-1 | Recall | Precision |
|---|---|---|---|
| 229 | 0.57 | 0.51 | 0.65 |
| 458 | 0.62 | 0.56 | 0.69 |
| 916 | **0.71** | **0.72** | **0.70** |

To assess the predictive capability of each note type in determining fall risk, we reported success rates in the test set, presenting the result in Table VIII alongside 95% binomial confidence intervals (CI). Our findings indicate that certain note types, such as Patient Summary, Outpatient, and PACU Assessment PowerForms, achieved success rates exceeding 90%, with 95% CI ranging from 71% to 98%. Each note type contains different kinds of information. For instance, patient

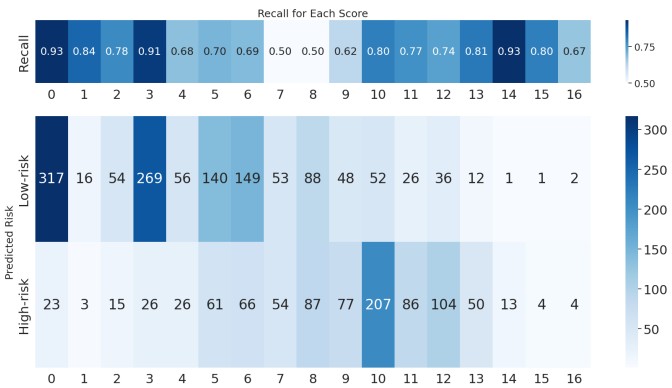

Fig. 6: Predicted risk for each Cummings Fall Assessment Score with recall values at the top.

summary outpatient notes often contain rehab-related information that may be more directly associated with fall risk. In contrast, admission notes include general details, such as skin color and temperature, which might be less predictive. The predictive power of these note types could vary across different hospital networks, depending on documentation practices and the specific patient populations. However, it's important to note that some note types had small sample sizes, so we can not conclude their informativeness. Nonetheless, these findings provide valuable insights into the potential value of each note type for fall risk assessment.

TABLE VIII: Success rates for note type that has more than 20 sample size on the test set with 95% confidence intervals (CI).

| Note Type | # of Notes | Success Rate | 95% CI |
|---|---|---|---|
| Patient Summary, Outpatient | 133 | 0.92 | [0.87,0.96] |
| PACU Assessment PowerForms | 22 | 0.91 | [0.71,0.98] |
| Outpatient Intake Information PowerForms | 633 | 0.89 | [0.86,0.91] |
| Postprocedure Assessment PowerForms | 75 | 0.80 | [0.69,0.88] |
| Patient Education PowerForms | 70 | 0.71 | [0.59,0.81] |
| Admission Assessment PowerForms | 482 | 0.69 | [0.65,0.73] |
| Pain Management Catheter Care PowerForms | 26 | 0.65 | [0.46,0.81] |
| Ongoing Shift Assessment PowerForms | 736 | 0.67 | [0.64,0.71] |
| Admission History PowerForms | 21 | 0.62 | [0.40,0.80] |

PACU: Post Anesthesia Care Unit.

## V. DISCUSSION

We leveraged unstructured EMR data to assess fall risk in children with CP, intending to reduce falls and improve patient outcomes. Despite the variability associated with CP patients, our findings demonstrated that BERT-based models can effectively perform fall risk assessment. By utilizing the available data and harnessing the power of generalized LMs, we are the first to show that CP fall risk assessment

management can be significantly enhanced, providing valuable support to clinicians in decision-making.

There are a few limitations of this work. Our study is confined to BERT-based models. For comparisons, we plan to explore other LM that showed promising results in recent years such as T5 [30], ClinicalT5 [31], BART [32]. Additionally, quality data can significantly enhance the performance of models. With more data availability we can increase model performance as shown in Table VII, by increasing 50% of subject number model performance increases 10%. This enhancement suggests that a larger dataset allows the model to learn more nuanced patterns and relationships within the data. The models can become better at generalizing from the training data to new unseen data, leading to improved accuracy and robustness in predictions. Moreover, our study is limited to fall risk assessment. Our goal is to extend predictive modeling using unstructured EMR in other prediction tasks that are important for children with CP, such as pain score prediction, and functional activity level evaluation.

We provide foundational baselines and experimented with both pre-trained and non-pre-trained for Clinical BioBERT, BioBERT, and BERT$_{BASE}$ in CP fall risk assessment. Our findings demonstrated that it is possible to leverage the capabilities of generalized LMs without the need for extensive domain-specific pre-training. This can significantly reduce the computational resources and time required for model pre-training. For instance, pre-training Clinical BioBERT took 17 - 18 days of runtime using GeForce GTX TITAN X 12 GB [18]. Our results indicate Clinical BioBERT and BERT$_{BASE}$ did not improve performance by pre-training, and BioBERT improved slightly. Further experimentation with more diverse datasets from multiple hospital networks is necessary to explore the benefits of pre-training, particularly with larger, more varied data sources. However, we showed utilization of existing pre-trained models for further tailored predictive modeling is both beneficial and cost-effective.

Moreover, to overcome LM's inability to process long text, we can make collected decisions from smaller pieces of longer notes. We utilized PWMV which can yield accurate decisions by giving more weight to individual samples with higher confidence [25]. Moreover, Meyen et al. introduced an equality effect parameter $\beta$; as $\beta$ gets larger that enhances the impact of high confidence segments relative to low confidence ones [26]. Without the equality parameter ($\beta = 1$), the PWMV treats each individual's confidence equally, whereas $\beta = 0$ can represent simple majority voting without weighing based on confidence. However, Meyen et al. demonstrated that using an equality parameter gives more preference to decisions supported by fewer individuals with very high confidence rather than a larger number of individuals with lower confidence. With PWMV we can make collective decisions, and model performances improved. This method is readily adaptable and can be applied, unlike approaches where increasing the context window model is expensive.

Utilizing cross-validated approaches can effectively reduce the uncertainty associated with evaluating fall risk. We achieved 0.71 F1 scores using BERT models in subject-wise testing. Subject-wise testing in a CP cohort is critical for providing the model's generalizability. It ensures that each person's unique challenges are addressed, facilitating better management of the condition and enhancing the quality of life for those affected. Fall risk changes over time, with an interplay between disease progression, age, and treatments. We have 622 out of 1144 subjects who changed over time between high and low risk according to the Cummings Fall Assessment Score. In this model, we are making risk assessment predictions when the notes were taken. For future work, we would like to make future predictions of fall risk, that is, anticipate changes in fall risk, but that is beyond the scope of this effort. Furthermore, we demonstrated that removing borderline samples improves model performance by 3%-4%. This indicates that models can effectively learn without the inclusion of hard samples, which may only have marginally differentiable features. These samples can be challenging not only to the model but also to clinicians, who might find it difficult to assess them accurately due to the lack of clear distinguishing characteristics. The presence of such samples can introduce noise and reduce the overall effectiveness of the model, highlighting the benefit of focusing on more distinct and well-labeled examples for training.

The models in our experiments showed signs of overfitting during training. For instance, the F1-scores for the fully fine-tuned Non-Pretrained Clinical BioBERT, BERT base, and BioBERT on the training set were 0.99, 0.98, and 0.98, respectively. To mitigate overfitting, we conducted further experiments by partially fine-tuning the Non-Pretrained Clinical BioBERT model. Instead of fine-tuning the entire model, we fine-tuned only the last encoder layer along with the added classification layers. This approach reduced overfitting, with the following results: on the training set, F1: 0.78, Precision: 0.87, Recall: 0.70, and on the test set, F1: 0.74, Precision: 0.80, Recall: 0.69. All the results mentioned above were obtained within the PWMV setup. Implementing more robust techniques to address overfitting could further enhance model performance.

Moreover, we show that certain types of samples possess greater predictive power than others. It provides clinicians with a clearer understanding of which types of data are the most critical for making decisions. By identifying these key data types, clinicians can focus their efforts on gathering and analyzing the most impactful information, thereby enhancing the accuracy of their assessments. In summary, our study leads to more reliable predictions, better-informed clinical decisions, and ultimately, improved outcomes for children with CP.

## VI. CONCLUSION

There is significant assistance needed in the current fall risk assessment in CP. This study suggests implementing BERT-based models for this population can significantly impact the evaluation process. By collecting more data, these models improve their accuracy and can offer a powerful tool for

integrating and analyzing diverse data sources, providing personalized and accurate risk predictions, and supporting more effective clinical decision-making. By addressing the current limitations in fall risk assessment, BERT-based models can improve outcomes and a higher quality of care for individuals with CP. Implementing these advanced models aligns with the growing emphasis on data-driven, personalized healthcare, and can potentially transform fall risk management in the CP population.

## FUNDING

This study was funded by Shriners Children's Clinical Research Grant 79148 (Improving Orthopedic Outcomes with Machine Learning, PI Krzak).

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
