# OpenReview forum: "Language Models for Fall Risk Assessment in Children with Cerebral Palsy using Electronic Medical Records"
_IEEE.org/EMBS/BHI/2024/Conference — IEEE BHI'24_

### Official Review · Reviewer_ECXJ · 2024-08-13
**Language Models for Fall Risk Assessment in Children with Cerebral Palsy using Electronic Medical Records**

**Overall Rating:** 7
**Confidence:** 3

**Other Quality Metrics:**

(a) Clarity of writing - Good
(b) Clinical Significance - Great
(c) Methodological Novelty - Good
(d) Experiments and Results - Great

**Questions For The Authors:**

(1) Would like to know the F1 score on the training and validation dataset to assess how much overfitting between training data and unseen EHR.
(2) Is fall risk constant or does it change over time? How might this impact your model and its downstream clinical implementation?

**Strengths:**

(1) The clinical problem is well-defined and it's importance is clearly expressed.
(2) The approach is methodologically sound and the use of unstructured EMR data is practical for the clinical application.
(3) The approach for pre-training and fine tuning may be generalizable to other clinical prediction tasks.
(4) Good comparison of multiple models and ablation studies to assess the impact of PWMV on classification performance

**Summary Of The Paper:**

The paper applies BERT-based language models, specifically Clinical BioBERT, BioBERT, and BERTBASE to predict fall risk in children with CP using unstructured EMR as input. The study introduces a Probability Weighted Majority Voting (PWMV) strategy to improve model performance by aggregating predictions from segmented clinical notes.

**Weaknesses:**

(1) It is unclear on what sample of data the 0.71 and 0.74 F1 scores are achieved on. The methods section states 80/20 train/test split, where the training data is further split into 80/20 for validation. Presumably, the test set of 228 subjects is not used in any part of the pre-training or fine-tuning of the models; however, this is not clear to the reader.
(2) Dataset appears to be from a single institution, while EHR documentation practices can vary by institution and EHR software used.

---

### Official Review · Reviewer_nz7H · 2024-08-13
**New application but with limited techinical novelty**

**Overall Rating:** 6
**Confidence:** 4

**Other Quality Metrics:**

a) Clarity of writing; great
(b) Clinical Significance; fair
(c) Methodological Novelty; poor
(d) Experiments and Results; fair

**Questions For The Authors:**

You mention 'certain types of samples possess greater predictive power than others', is it because of certain exams? Can you please provide a few examples of such samples? Do you think the same type of samples will still hold if at other hospitals?

**Strengths:**

well-written paper with a novel application

**Summary Of The Paper:**

This paper uses the Language Model to evaluate fall risk using electronic medical records. The authors explored the impact of pre-training and fine-tuning on performance. They also used a voting-based method to handle the extreme-long context info.

**Weaknesses:**

Technical novelty is limited, pre-training and fine-tuning are both mature techniques.

Also as other reviewers have mentioned, given the single source of the dataset (only from one hospital), the conclusion (pre-training is not necessary) is less convincing.

---

### Official Review · Reviewer_s11h · 2024-08-14
**Important problem and valuable analysis**

**Overall Rating:** 7
**Confidence:** 3

**Other Quality Metrics:**

Clarity - good
Clinical significance - great
Methodological novelty - fair
Experiments and Results - good

**Questions For The Authors:**

None

**Strengths:**

Most of the previous works in fall detection usually work with elderly people but this paper covers a different important demographic group. The model clearly describes the methodology, the dataset, and the results with helpful figures. The paper presents the performances of their methods for different note types, which can be beneficial for future researchers and clinicians to decide how to maintain clinical notes more efficiently for children with cerebral palsy (CP).

**Summary Of The Paper:**

The paper proposes a solution that can assess the risk of falls in children with cerebral palsy. It uses BioBERT, Clinical BioBERT, and BERT_base to predict fall assessment risk. The method uses clinical notes to categorize patients into high-risk and low-risk groups.

**Weaknesses:**

None

---

### Decision · Program_Chairs · 2024-09-23

Accept